# Evaluation of the Relationship between Fractional Exhaled Nitric Oxide (FeNO) with Indoor PM_10_, PM_2.5_ and NO_2_ in Suburban and Urban Schools

**DOI:** 10.3390/ijerph19084580

**Published:** 2022-04-11

**Authors:** Khairul Nizam Mohd Isa, Juliana Jalaludin, Saliza Mohd Elias, Norlen Mohamed, Jamal Hisham Hashim, Zailina Hashim

**Affiliations:** 1Department of Environmental and Occupational Health, Faculty of Medicine and Health Sciences, Universiti Putra Malaysia, UPM, Serdang 43400, Selangor, Malaysia; khairulnizamm@unikl.edu.my (K.N.M.I.); saliza_me@upm.edu.my (S.M.E.); drzhashim@gmail.com (Z.H.); 2Environmental Health Research Cluster (EHRc), Environmental Healthcare Section, Institute of Medical Science Technology, Universiti Kuala Lumpur, Kajang 43000, Selangor, Malaysia; 3Environmental Health Unit, Level 2, E3, Disease Control Division, Ministry of Health, Putrajaya 62590, Wilayah Persekutuan Putrajaya, Malaysia; norlen.mohamed@moh.gov.my; 4Department of Health Sciences, Faculty of Engineering and Life Science, Universiti Selangor, Shah Alam Campus, Seksyen 7, Shah Alam 40000, Selangor, Malaysia; jamalhas@hotmail.com

**Keywords:** FeNO, school children, urban, suburban, indoor pollutants

## Abstract

Numerous epidemiological studies have evaluated the association of fractional exhaled nitric oxide (FeNO) and indoor air pollutants, but limited information available of the risks between schools located in suburban and urban areas. We therefore investigated the association of FeNO levels with indoor particulate matter (PM_10_ and PM_2.5_), and nitrogen dioxide (NO_2_) exposure in suburban and urban school areas. A comparative cross-sectional study was undertaken among secondary school students in eight schools located in the suburban and urban areas in the district of Hulu Langat, Selangor, Malaysia. A total of 470 school children (aged 14 years old) were randomly selected, their FeNO levels were measured, and allergic skin prick tests were conducted. The PM_2.5_, PM_10_, NO_2_, and carbon dioxide (CO_2_), temperature, and relative humidity were measured inside the classrooms. We found that the median of FeNO in the school children from urban areas (22.0 ppb, IQR = 32.0) were slightly higher as compared to the suburban group (19.5 ppb, IQR = 24.0). After adjustment of potential confounders, the two-level hierarchical multiple logistic regression models showed that the concentrations of PM_2.5_ were significantly associated with elevated of FeNO (>20 ppb) in school children from suburban (OR = 1.42, 95% CI = 1.17–1.72) and urban (OR = 1.30, 95% CI = 1.10–1.91) areas. Despite the concentrations of NO_2_ being below the local and international recommendation guidelines, NO_2_ was found to be significantly associated with the elevated FeNO levels among school children from suburban areas (OR = 1.11, 95% CI = 1.06–1.17). The findings of this study support the evidence of indoor pollutants in the school micro-environment associated with FeNO levels among school children from suburban and urban areas.

## 1. Introduction

Many health-related assessments on the indoor air pollutants in the school settings have been conducted. The evaluations of risk assessment on school children are particularly critical, because schools represent a distinctive and vital micro-environment. Consistent results supporting those higher risks of respiratory allergic diseases were significantly associated with poor indoor air quality setting [1,2]. Asthma is the most common illness of childhood, affecting 339 million and resulting in the deaths of 13,909 children globally in 2016 [3]. Additionally, Achakulwisut et al. [4] estimated that 4.0 million of new paediatric asthma cases at global levels could be linked to annual NO_2_ pollution and 64.0% of cases occur in urban centres.

On another note, the concentrations of particulate matter (PM), black carbon, nitrogen dioxide (NO_2_) and ozone (O_3_) have always shown significant variation between urban (cities and megacities) and nonurban (suburban, rural, remote) areas [5]. For example, studies have shown that the concentration of O_3_ is usually higher in suburban areas, while the concentration of NO_2_ is generally opposite to O_3_, which depends heavily on the road traffic [6,7]. Generally, air pollution in the urban environment can be associated with anthropogenic sources as well as motor vehicle exhaust and industrial emissions [8]. Additionally, the greater density of road traffic and buildings in urban areas may intensify the generation of urban heat island and urban pollution island phenomena than in nonurban areas [9]. These variations can have substantial impacts on vulnerable groups, including children and the elderly. Furthermore, previous research works have focused on urban areas with high pollution and emission levels. There is little data establishing the differences between exposure in urban and suburban school environments. Moreover, the exposure effects of air pollutants exposure in suburban areas cannot be overlooked. Hence, further evidence from epidemiological studies is needed to obtain more comprehensive knowledge about the impacts of the major air pollutants (PM_10_, PM_2.5_, NO_2_) in school micro-environments in both, urban and suburban areas.

In recent years, the evaluation of airway inflammation which is associated with air pollutants had been made very simplistically and noninvasively with direct measurement of fractional exhaled nitric oxide concentration in exhaled breath (FeNO) using chemiluminescence analyser [10]. Nitric oxide (NO) is the most widely used biomarker and found to be directly correlated with the severity of airway inflammation [11]. A growing body of evidence shows that short term exposure to the major air pollutants, including PM_10_, PM_2.5_, NO_2_ and O_3_ are linked to changes in FeNO levels [12,13,14]. Nevertheless, the results are inconsistent. For instance, Gong et al. [12] and Prapamontol et al. [15] have demonstrated that ultrafine particles (UFP) and PM_10_, respectively, measured in urban school environments, were negatively associated with FeNO levels, due to other independent factors including timing of exposure and prevailing weather conditions [16]. Moreover, due to lack of research in analysing the comparison of effects of these major pollutants on school children between urban and suburban areas, it is impossible to further characterise the strategies for air pollution controls by the local authorities.

For these reasons, we conducted the present study to investigate the association of FeNO with indoor PM_2.5_, PM_10_ and NO_2_ exposure in urban and suburban school areas. This is critical in order to better understand the significant variations of indoor pollutants exposure that may contribute to respiratory allergic diseases among school children.

## 2. Materials and Methods

### 2.1. Study Population

This comparative cross-sectional study was conducted among school children aged 14 years old from secondary schools in the Hulu Langat district, Selangor, Malaysia. The Hulu Langat district features an urban sprawl from rapid urbanisation of Kuala Lumpur and Putrajaya, where there are massive construction development projects for the industrial and property estates [17]. A stratified random sampling design was followed for selection of schools. The school areas were classified as urban and suburban based on the locale classification of ecological measures by the Ministry of Education, Malaysia. The number of schools selected was defined based on the sample size of school children in relation to respiratory symptoms [18]. A total of eight schools were used to give satisfactory confidence. Thus, four single session schools (afternoon) were randomly selected from each school area. Sample size calculation was adjusted for stratification sampling using a design effect of 1.1 and 0.02 was considered as an anticipated value for inter-cluster correlation (ICC) based on a comparable study conducted in children [19]. A total of 470 school children were recruited for this study. They were randomly selected from four classrooms in each school. In the selection process, school children who have been attending the same school since January 2017 (or more than 18 months) and obtained written consent from parents or legal guardians with the addition of their own assent were included. On the other hand, school children with concomitant heart diseases and severe asthma conditions were excluded. We also excluded school children who had incomplete data and their respective residential address outside of school area. The clinical assessment and indoor air monitoring were carried out at the same time from August until November 2018 and in early February 2019. This study was approved by the Ethics Committee for Research Involving Human Subjects Universiti Putra Malaysia (JKEUPM) (JKEUPM-2018-189).

### 2.2. Clinical Assessment

Information on demographic characteristics, doctor’s diagnosed asthma, current asthma medication and any asthma attack during the last 12 months were collected by self-administrative questionnaire. The questionnaire was adapted from the International Study of Asthma and Allergies in Childhood (ISAAC), the European Community Respiratory Health Survey (ECRHS) and previous studies [20,21]. This information was verified during face-to-face interviews and telephone calls with the children’s respective guardians.

Airway inflammation was assessed by measuring the fractional exhaled nitric oxide (FeNO) using chemiluminescence analyser (NIOX VERO, Circassia, Sweden) with a detection limit and accuracy of 5–300 ppb and ±5 ppb, respectively. This analyser has visual and audio signals guide the school children to achieve the desired expiratory flow of 50 mL/s in six to ten seconds. Samples of exhaled breath were taken in accordance with the standard and as recommended by the manufacturer [22]. A single-breath technique was used, and this procedure was repeated at least twice to obtain an average value. School children were instructed to avoid eating and drinking for at least an hour before the FeNO assessment. To exclude errors related to the time of sampling, all the FeNO assessment was performed in the afternoon between 1.00 PM and 6.00 PM by trained enumerators.

All school children with guardian consent and own assent underwent skin prick test with five common allergens purchased from ALK-Abelló, (Madrid, Spain): *Dermatophagoides pteronyssinus, Dermatophagoides farina, Cladosporium herbarum, Alternaria alternate*, and *Felis domesticus*, also Histamine (10 mg/mL) and glycerol-saline were used as the positive and negative controls, respectively. The procedures of skin prick test were carried out by trained medical assistants and in accordance with the Australasian Society of Clinical Immunology and Allergy guidelines. The reaction was measured after 15 min by measuring the wheal diameter. The allergen’s wheal diameter of 3 mm was considered as a positive control. Atopy was defined as a significant positive skin prick test to at least one of the applied allergens [23].

### 2.3. Assessment of School Environment

Four classrooms in each school were randomly selected and inspected for signs of dampness or mould growth. Indoor temperature (°C), relative humidity (%), CO_2_ (ppm) were monitored in the classrooms by using Q-TrakTM IAQ monitor (Model 7565 TSI Incorporated, Shoreview, MN, USA) with the average log interval values over one minute. The accuracy of this device on temperature, relative humidity, and CO_2_ are ±0.6 °C, ±3%, and ±50 ppm, respectively. The concentrations of PM_10_ and PM_2.5_ (µg/m^3^) were measured using two separate units of Dust-Trak monitor (Model 8532 TSI Incorporated, Shoreview, MN, USA) at a sampling rate of 1.7 L/min and a detection limit of 0.001–150 mg/m^3^. All of these samplers were always placed one metre above floor level and one metre away from the school children in the centre of the classrooms.

In each school, a total of four hours of measurements were collected for all of these parameters during the learning session between 1.00 PM and 6.00 PM to give more realistic exposure estimation and has been previously described [14,24,25,26]. For measurement of NO_2_ (µg/m^3^), the IVL diffusion samplers (IVL, Goteborg, Sweden) were used with the limit of detection (LOD) of 0.5 µg/m^3^ and 10.0% of measurement uncertainty [27]. This passive diffuser sampler was used to determine the average concentration of NO_2_ in the air for a week.

### 2.4. Statistical Analysis

Descriptive test analysis was performed by the Statistical Package for Social Science (SPSS) 25.0. We used log transformed data of indoor parameters to improve normality in the regression analysis. The chi-squared test was used to compare the school children’s characteristics with respect to school areas, while statistical comparisons for FeNO levels and indoor pollutant parameters were made using the Mann–Whitney test. The two-level hierarchical multiple logistic regression (school and school children) was performed using the Stata/MP 15.1 (StataCorp LLC, College Station, TX, USA) to evaluate the association between categorised FeNO levels (normal vs. elevated level; >20 ppb) as the dependent variable and indoor pollutant parameters, controlling for gender, atopy, doctor-diagnosed asthma, weight, height, smoking status, parental asthma/allergy and family member smoking status [28,29]. A FeNO level above 20 ppb was considered as an elevated level, as recommended by the American Thoracic Society (ATS) guidelines for children [30]. We analysed the association models for each school area separately. Additionally, we incorporated the sampling weights [31] with the formula recommended by Foy [32] in the multivariate analysis stage to compensate for unequal probability of selection at classroom and school children levels. All tests were 2-tailed, and a *p*-value of less than 0.05 was considered significant.

## 3. Results

### 3.1. Personal Characteristic and FeNO Levels

Table 1 summarises the personal characteristics of school children involved in this study. Two hundred (42.6%) of the school children were from suburban and 270 (57.4%) from urban school areas, of which 61.3% were female. The overall prevalence of doctor-diagnosed asthma was 10.6%. Moreover, a total of 57.7% of school children had been sensitised to at least one of the allergens tested. However, the prevalence of doctor-diagnosed asthma and atopy did not significantly differ by the school areas. A total of 56.1% of the school children from urban areas were more likely to be exposed to secondhand tobacco smoke (SHS) at home. The prevalence of allergy and/or asthma was slightly higher in the parents from urban (52.3%) than suburban (47.7%) areas.

The level of FeNO was nearly the same among school children in suburban (median = 19.5, IQR = 24.0) and urban areas (median = 22.0, IQR = 32.0) (Figure 1). Nevertheless, comparison analysis revealed that the FeNO levels were significantly different by gender among school children from urban school areas (*p* < 0.05). Additionally, there were statistically significant higher levels of FeNO in doctor-diagnosed asthma and atopy school children from both urban and suburban school areas (*p* < 0.001) (Table 2). In particular, 71.4% and 72.4% of school children with doctor-diagnosed asthma conditions from suburban and urban areas, respectively, were significantly recorded with elevated FeNO levels (*p* < 0.05). Similarly, a higher proportion of school children with elevated FeNO levels were observed among atopic groups from both areas (*p* < 0.001).

### 3.2. Classroom Inspection and Indoor Environmental Parameters

Generally, all classrooms were equipped with three ceiling fans, naturally ventilated and designed with glass jalousie window panes on both sides of the wall. The floor surface was finished with concrete. None of the classrooms had signs of dampness or mould growth. About 25 classrooms were occupied with plastic tables and the rest of the classrooms were wooden tables. Only four classrooms used wooden chairs, while the others used plastic chairs. There were bookshelves, whiteboard, and soft boards in every classroom.

From the comparison analysis, there were significant differences (*p* < 0.001) in all indoor parameters observed between schools located in urban and suburban areas, except the concentration of CO_2_ (*p* > 0.05). Generally, the temperature level, the concentrations of NO_2_, PM_10_ and PM_2.5_ recorded in schools located in urban areas were moderately higher than in suburban areas (Table 3).

### 3.3. Association of Indoor Pollutants with FeNO Levels

The results of two-level hierarchical multiple logistic regression models for school children from suburban and urban school areas are shown in Table 4. After controlling gender, atopy, doctor-diagnosed asthma, weight, height, parental asthma/allergy and smoking status, we found only the concentrations of PM_2.5_ were significantly associated (OR = 1.30, 95% CI = 1.10–1.91) with elevated of FeNO levels in school children from urban areas (*p* < 0.05). In the second model, the concentration of NO_2_ was found to be significantly associated with the elevated FeNO levels with an odd 1.11 (95% CI = 1.06–1.17) compared with the normal FeNO group in school children from suburban areas. Notably, the same model also shows that the concentration of PM_2.5_ was positively associated with the elevation of FeNO levels with the odd 1.42 (95% CI = 1.17–1.72).

## 4. Discussion

This comparative cross-sectional study explored the association of FeNO levels with indoor air pollutants exposure among 470 school children in urban and suburban school areas. Further, this study demonstrated significant associations of indoor PM_2.5_ and NO_2_ concentrations with FeNO levels among school children in both school areas. To the best of our knowledge, our study is one of very few studies that provide epidemiological evidence for the link between FeNO levels and indoor pollutants measured in school’s micro-environment in suburban and urban areas. Our findings also add to the scant evidence about the adverse effects of PM_2.5_ and NO_2_ in school micro-environment on airway inflammation in Southeast Asia.

In this current study, we found that the prevalence of doctor-diagnosed asthma and atopy was 10.6% and 57.7%, respectively. Nevertheless, the latest prevalence of doctor-diagnosed asthma among children aged 13 to 14 years old reported from local studies in Terengganu (Malaysia) and Penang (Malaysia) were 8.4% [21] and 10.3% [33], respectively. Compared to other studies in Southeast Asia, the prevalence of doctor-diagnosed asthma in Bangkok (Thailand), Singapore, and Surabaya (Indonesia) was 8.8% [34], 10.0% [35] and 6.8% [36], respectively. Therefore, there seems to be an indication that asthma prevalence is on the rise in Malaysia. Overall, in most countries, an increased prevalence of asthma has been documented compared to the past century [37]. Moreover, according to Sembajwe et al. [38], the prevalence of doctor-diagnosed asthma across the world regions was reflected by the national incomes. Likewise, the prevalence of atopy identified by previous studies also shows lower percentages than this current study. Previous studies in Terengganu (Malaysia), Surabaya (Indonesia) and South Korea found that the prevalence of atopy among similar age group were 40.3% [21], 29.0% [36] and 27.3% [39], respectively.

Furthermore, we observed that the prevalence of doctor-diagnosed asthma and atopy were more pronounced in school children from urban areas than suburban areas. These findings were consistent with those of other studies conducted among Chinese children [40] and Korean children [41]. This also accords with the extensive body of evidence on the role of atopy as a major risk factor for asthma, rhinitis and eczema in children [42].

With regard to the comparison analysis between urban and suburban areas, results from this current study shows that the FeNO levels were not statistically different between these groups. This result is supported by the previous comparative studies conducted in Selangor (Malaysia; urban/suburban), Terengganu, (Malaysia; urban/rural) and Bilthoven (Netherlands; urban/suburban) [43,44,45]. Thus, this indicates that school children from urban and suburban areas have a normal FeNO range, 20–35 ppb, as recommended by the American Thoracic Society (ATS) [30]. However, further comparison analysis provided additional evidence that the proportion of elevated FeNO levels were significantly higher in school children who had diagnosed asthma and atopy from both areas, suburban and urban, that could reflect a higher degree of airway inflammation. These results are consistent with the other studies and reviews reported [29,46,47]. A possible explanation for this might be that urban environmental irritants stimulate the development of allergic sensitisation [41].

Overall, we found that the concentrations of PM_10_, PM_2.5_, NO_2_ and CO_2_ measured inside the classrooms were below the guideline limits set by the WHO guidelines [3], the National Ambient Air Quality Standard by USEPA [48], the new Malaysian Ambient Air Quality Standard 2018 Interim Target-2 [49] and the Industrial Code of Practice on Indoor Air Quality (ICOP-IAQ) 2010 [50]. This was due to inflow of outdoor air through the jalousie window panes on both sides of the wall and adequate ceiling fans which could have made the natural ventilation effective [51,52].

In the multivariate analysis, we found that the PM_2.5_ exposures were positively associated with elevated FeNO levels among school children from both areas, suburban and urban. In accordance with previous results, Shang et al. [53] reported that the ambient PM_2.5_ measured in urban areas has strong association with FeNO levels. Another epidemiological study conducted in Taiyuan City, China also reported that only the concentration of PM_2.5_ measured inside the classroom was significantly associated with FeNO levels [54]. These findings were consistent with the review articles by Qibin et al. [55] and Chen et al. [56]. They indicated that PM_2.5_ can directly produce a significant number of free oxygen radicals, which lead to activation of inducible NOS (iNOS) expression and prolonged release of high amounts of NO in the airways. This has been confirmed by Long et al. [57] in their in vitro study, reported recently.

Another important finding in this current study was the significant association of the increased NO_2_ concentration with elevated FeNO levels among school children from suburban areas. This finding support the results in the previous studies conducted by Olaniyan et al. [58] and Kamaruddin et al. [11], who also indicated that concentrations of NO_2_ were positively associated with a three-fold and five-fold increased risk of elevated FeNO levels among school children aged 9–11 years old in the Western Cape Province of South Africa and Terengganu, Malaysia, respectively. Nevertheless, we fail to find a significant association between elevated FeNO levels with concentration of NO_2_ in urban school areas, which is in agreement with those obtained by Gaffin et al. [59] and Carlsen et al. [60]. This result may be explained by the fact that differences in study design or characteristics of school children (history of asthma, smoking status) or different methods of assessing the FeNO levels [61]. The underlying mechanisms of NO_2_ influences the FeNO levels are unclear [62]. However, there is some evidence reported that the FeNO may be modulated by DNA methylation which is involved in the arginase–nitric oxide synthase pathway [16,56,62]. Interestingly, a longitudinal study conducted by Zhang et al. [54] and Jiang et al. [62] reported that short-term exposure to NO_2_ and PM_2.5_ significantly decreased the NOS2A and increased the ARG2 methylation. Therefore, their findings provide insights to enhance our understanding of the NO_2_ pathophysiology mechanisms, for which further studies are therefore suggested.

Some limitations of this study should be noted. First, the exposure assessment was collected over an average of one week for NO_2_, while the other indoor pollutants were collected with an average of a four-hour time frame in each school, which might have introduced exposure measurement errors in evaluating the multiple-pollutants effects. Nevertheless, we attempted to specify almost 80% of the exposure time frame during the school hours. Despite this, we found compelling evidence that linked the PM_2.5_ and NO_2_ exposure to FeNO levels. Second, the measurement of air pollutants parameters could be improved by incorporating outdoor air quality monitoring and at-home indoor settings, which potentially contribute to the respiratory outcomes. However, the measurements for indoor and outdoor parameters are expected to be constant throughout the year. This is supported by several studies conducted across Peninsular Malaysia [63,64]. Third, the other possible confounding factors in the home environment such as dampness/mould, furry pets, environmental tobacco smoke (ETS), residential materials and redecoration activities were not comprehensively addressed in this current study. Furthermore, it should also be noted that the questionnaire used was based on self-reporting and may have introduced recall bias. To address this limitation, a face-to-face interview session was conducted following the completion of the questionnaire and telephone call, with their respective parents, to verify the self-reporting information. Another limitation of this study is that the classification of atopy was based on a small number of allergens, which may have underestimated the prevalence of atopy. Finally, through the nature of the study design, the cross-sectional study design utilised here preludes establishing causation inferences.

## 5. Conclusions

In summary, the exposure levels of indoor pollutants in the school environment in this current study are in accordance with the WHO guidelines, the National Ambient Air Quality Standard by USEPA, the ICOP-IAQ 2010 and the new Malaysian Ambient Air Quality Standard 2018 Interim Target-2. Moreover, this study suggested that there is an independent relationship between PM_2.5_ and NO_2_, although less than in the existing guidelines, that adversely affects the FeNO levels in school children from suburban and urban areas. Therefore, the results from this current study have provided additional evidence to reinforce the effects of indoor pollutants in both school environments on the respiratory outcomes among school children. This is an important finding and indicates that further intervention studies are needed to identify the most effective mechanisms to comprehensively reduce the risks of indoor pollutants exposure in school micro-environment settings. Taking into consideration the school buildings in Malaysia, which are designed with a natural ventilation system and often situated nearby heavy traffic roads, there is also a need to formulate policies to control air pollution in urban and suburban areas, and to strengthen the prevention measures. For example, the implementation of strategies such as locating buildings away from high density traffic roads and creation of asthma-friendly school programs are recommended.

## Figures and Tables

**Figure 1 ijerph-19-04580-f001:**
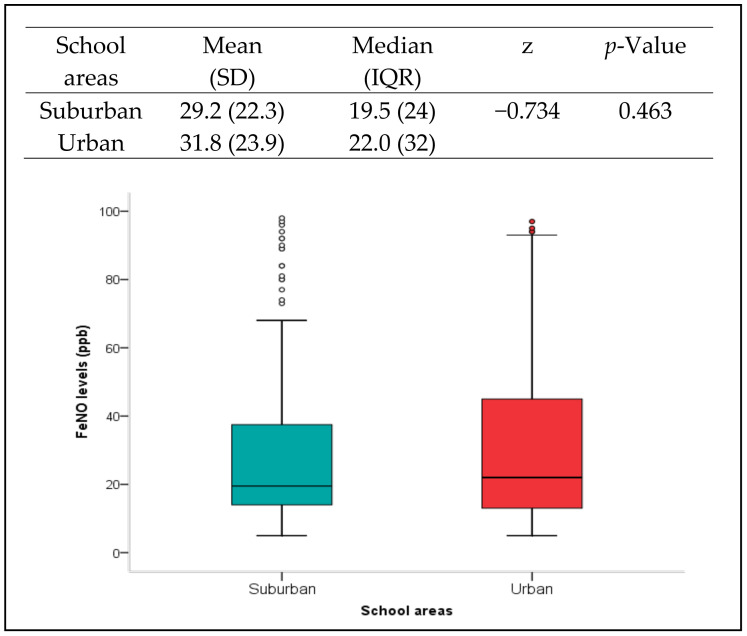
Summary of FeNO levels in school children by school areas, suburban and urban. The *p*-value refers to the comparison analysis using Mann–Whitney test. Box plots showing median values (presented by a horizontal line inside the box) and percentiles ranges (10th, 25th, 75th and 90th) of FeNO levels in school children by school areas. Circles represent outliers.

**Table 1 ijerph-19-04580-t001:** Characteristics of school children in suburban and urban school areas.

Characteristics	Overall (*n* = 470)	Suburban (*n* = 200)	Urban (*n* = 270)	*p*-Value
Gender				
Male	182 (38.7)	66 (36.3)	116 (63.7)	0.028 *
Female	288 (61.3)	134 (46.5)	154 (53.5)	
Ethnicity				
Malay	408 (86.8)	195 (47.8)	213 (52.2)	<0.001 **
Non-Malay	62 (13.2)	5 (8.1)	57 (91.9)	
Doctor-diagnosed asthma				
Yes	50 (10.6)	21 (42.0)	29 (58.0)	0.933
No	420 (89.4)	179 (42.6)	241 (57.4)	
Atopic				
Yes	271 (57.7)	110 (40.6)	161 (59.4)	0.315
No	199 (42.3)	90 (45.2)	109 (54.8)	
Parental allergy/asthma				
Yes	155 (33.0)	74 (47.7)	81 (52.3)	0.110
No	315 (67.0)	126 (40.0)	189 (60.0)	
Smoking				
Yes	30 (6.4)	9 (30.0)	21 (70.0)	0.151
No	440 (93.6)	191 (43.4)	249 (56.6)	
Parental/sibling smoking				
Yes	285 (60.6)	125 (43.9)	160 (56.1)	0.477
No	185 (39.4)	75 (40.5)	110 (59.5)	

* *p* < 0.05; ** *p* < 0.001.

**Table 2 ijerph-19-04580-t002:** Difference of FeNO levels (ppb) and prevalence of elevated FeNO levels (>20 ppb) between school children from suburban and urban areas by their characteristics.

Characteristics	Overall (N = 470) Median (IQR)	Suburban			Urban		
		Normal *n* (%)	Elevated *n* (%)	*p*-Value	Normal *n* (%)	Elevated *n* (%)	*p*-Value
Gender							
Male	26 (31)	32 (48.5)	34 (51.5)	0.549	42 (36.2)	74 (63.8)	0.004 *
Female	19 (26)	71 (53.0)	63 (47.0)		83 (53.9)	71 (46.1)	
Ethnicity							
Malay	21 (28)	99 (50.8)	96 (49.2)	0.402	99 (46.5)	114 (53.5)	0.907
Non-Malay	21 (25)	4 (80.0)	1 (20.0)		26 (45.6)	31 (54.4)	
Doctor-diagnosed asthma							
Yes	56 (63)	6 (28.6)	15 (71.4)	0.026 *	8 (27.6)	21 (72.4)	0.032 *
No	20 (23)	97 (54.2)	82 (45.8)		117 (48.5)	124 (51.5)	
Atopic							
Yes	32 (38)	38 (34.5)	72 (65.5)	<0.001 **	50 (31.1)	111 (68.9)	<0.001 **
No	16 (12)	65 (72.2)	25 (27.8)		75 (68.8)	34 (31.2)	
Parental allergy/asthma							
Yes	22 (30)	38 (51.4)	36 (48.6)	0.974	34 (42.0)	47 (58.0)	0.351
No	21 (27)	65 (51.6)	61 (48.4)		91 (48.1)	98 (51.9)	
Smoking							
Yes	19.5 (40)	4 (44.4)	5 (55.6)	0.927	12 (57.1)	9 (42.9)	0.299
No	21 (27)	99 (51.8)	92 (48.2)		113 (45.4)	136 (54.6)	
Parental/sibling smoking							
Yes	21 (28)	61 (48.8)	64 (51.2)	0.324	78 (48.8)	82 (51.2)	0.329
No	22 (29)	42 (56.0)	33 (44.0)		47 (42.7)	63 (57.3)	

* *p* < 0.05; ** *p* < 0.001.

**Table 3 ijerph-19-04580-t003:** Comparison of indoor air pollutants between suburban and urban areas.

Parameter	Suburban Median (IQR)	Urban Median (IQR)	*p*-Value	Reference
	*n* = 16	*n* = 16		
Temperature (°C)	27.0 (1.0)	29.0 (2.0)	<0.001 **	23–26 ^a^
Relative humidity (%)	80.4 (7.5)	74.6 (9.5)	<0.001 **	40–70 ^a^
CO_2_ (ppm)	456.0 (27.0)	452.0 (33.0)	0.068	<1000 ^a,b,c,d^
NO_2_ (µg/m^3^)	20.0 (29.0)	32.0 (5.0)	<0.001 **	200 ^b^, 100 ppb ^c^, 75 ^d^
PM_2.5_ (µg/m^3^)	21.9 (2.1)	24.3 (2.5)	<0.001 **	25 ^b^, 35 ^c^, 50 ^d^
PM_10_ (µg/m^3^)	36.7 (2.7)	41.0 (7.3)	<0.001 **	50 ^b^, 150 ^c^, 120 ^d^

N = 32. IQR = interquartile range. ** *p* < 0.001. ^a^ Industrial Code of Practice on Indoor Air Quality (ICOP-IAQ) 2010. ^b^ World Health Organization (WHO) guideline. ^c^ The National Ambient Air Quality Standard by USEPA. ^d^ The new Malaysian Ambient Air Quality Standard 2018 Interim Target-2.

**Table 4 ijerph-19-04580-t004:** Association of elevated FeNO levels (>20 ppb) in school children from suburban and urban areas with indoor air pollutants.

Parameter	Suburban (*n* = 200)		Urban (*n* = 270)	
	OR	95% CI	OR	95% CI
Temperature (°C)	0.84	(0.31–2.27)	0.99	(0.75–1.32)
Relative humidity (%)	0.97	(0.87–1.08)	0.94	(0.89–0.96)
NO_2_ (µg/m^3^)	1.11	(1.06–1.17) **	1.02	(0.97–1.06)
CO_2_ (ppm)	1.02	(0.99–1.05)	1.00	(0.98–1.03)
PM_2.5_ (µg/m^3^)	1.42	(1.17–1.72) *	1.30	(1.10–1.91) *
PM_10_ (µg/m^3^)	0.89	(0.75–1.08)	1.09	(0.98–1.20)

* *p* < 0.05; ** *p* < 0.001. CI = confidence interval. OR calculated for 10 µg/m^3^ increase in concentration of NO_2_, PM_2.5_, PM_10_. OR calculated for 100 ppm increase in concentration of CO_2_. OR (OR = odds ratio) was calculated by two-level hierarchical multiple logistic regression.

## Data Availability

The data presented in this study are available on request from the first author. The data are not publicly available due to privacy.

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
