# Peer review of "Evaluation of the Relationship between Fractional Exhaled Nitric Oxide (FeNO) with Indoor PM10, PM2.5 and NO2 in Suburban and Urban Schools"

_ijerph, 2022, doi:10.3390/ijerph19084580_

Round 1
Reviewer 1 Report
This study investigated the association between indoor air pollution and airway inflammation in 470 children from randomly-selected 4 classes of 8 schools located in urban and suburban areas, Hulu Langat district, Malaysia. Indoor air pollution of PM2.5, PM10, and NO2 were sampled in each classroom, while airway inflammation was measured as fractional exhaled nitric oxide (FeNO). Using binary FeNO, authors applied the hierarchical logistic model adjusting for gender, atopy, doctor-diagnosed asthma, weight, height, smoking status, parental asthma/allergy, and family member smoking status. Findings showed significantly positive odd ratios of FeNO for NO2 in suburban areas and for PM2.5 in urban and suburban areas.
Major comments:
This study focused on an important population of children and provided interesting findings to clarify mechanism for the health effect of indoor air pollution at schools based on locally collected data. However, the motivation and research methods should be further clarified for publication. First of all, it does not seem to be clear what this study aimed to. I agree that it is important to study airway inflammation and air pollution in school children. However, there are many previous findings as described by authors. What is the novelty of this study? In the Introduction, authors mentioned previous findings are inconsistent without providing references and describing how those inconsistency are. Does this study aim to provide new evidence in Malaysia, as authors mentioned studies of air pollution in Malaysia is lacking in the Introduction? If so, how can the addition of the local finding contribute to international scientific society? Would the novelty be the inclusion of both urban and suburban areas which were split through the study? Then, why do we need to look at suburban and urban areas separately? Were there few studies in suburban or urban areas, or do we need a comparison between the two areas? In the Introduction, authors described the severity of air pollution in urban areas but did not provide more in terms of the motivation of dividing into urban and suburban areas. In the Methods section, authors mentioned all eight schools were located in urban sprawl without further information on how these 8 schools or students were classified to urban and suburban areas. Were these urban and suburban areas defined as schools or children’s residences? If it’s school, would it be possible that some children live in other types of areas from school areas? Authors may want to clarify what they specifically aimed to and why we should focus on those aims. Secondly, the temporal resolution does not sound clear for me, although this is the key information on temporally-sensitive exposure and outcome such as air pollution and FeNO. In the methods, authors told ‘four hours of measurements for all parameters during the learning session’, ‘CO2 within an hour’, and ‘average concentration of NO2 in the air for a week’. All of these descriptions provided different temporal scales which do not make sense. I googled the three cited papers for exposure assessment, but it is difficult to find the information either. One paper provides only abstract, the second paper does not look related to air pollution, and the last one is not even found by using the title. If air pollution and/or FeNO are collected at different times and days across 480 children, these times and days may behave as critical confounders and need to be accounted for in statistical analyses. Also, authors mentioned that they ‘incorporated the sampling weights in the multivariate analysis stage’. However, they only mentioned random sampling of classes without introducing any sampling weight over the Methods. Authors may want to provide more details on what this weight means and how they incorporated in analyses. Lastly, the odds ratio and 95% CI of feNO for PM2.5 from urban school students in Table 4 are 1.03 and 1.10-1.91 which do not look correct.
Author Response
We thank the reviewer for your valuable input and comments on the manuscript. We hope that our reply to each of your comments/suggestions finds your approval. We additionally uploaded the response letter as a file.
Thank you.

Reviewer 2 Report
The current manuscript paper entitled “Evaluation of the Relationship between Airway Inflammation with Indoor PM10, PM2.5 and NO2 in Suburban and Urban Schools” aimed to investigate the associations between a marker of airway inflammation (FeNO) and indoor particulate matter (PM10 and PM2.5), and nitrogen dioxide (NO2) exposure, in order to understand the role of risks of air pollution between schools located in suburban and urban areas and its consequences in respiratory health among schools' children.
General comments:
- I would suggest writing "FeNO" as a marker of airway inflammation, NOT "airway inflammation".
- Most of the references are recent, however, too many references (59) were included for an original paper.
There are some other comments which need to be addressed:
Abstract:
- The "IQR" was reported wrongly and should be a value between 2 numbers.
- The sampling technique was not clearly reported.
- Line 30: ...... (OR= 1.03, 95% CI= 1.10-1.91) -----> the value of this "CI" needs to be revised, as the "OR" value should lie between the range value of the "CI".
Introduction:
- There is a lack of cohesion between the statements and the paragraphs in the "introduction".
- What was the rationale for the selection of only 3 air pollutants (PM2.5, PM10, NO2)? what about (O3, SO2, VOCs, ... etc)?
Materials and Methods:
- Again, the authors need to clarify what was the used sampling technique and methods for the selected schools?
- More explanation about the SPT procedure is needed here? (positive /negative controls? the wheel size for defining a positive SPT?
Results:
- Again, there was incorrect reporting of the "IQR", as it should be a range value of 2 numbers.
- TABLE 2: ---- what is the p-value for the differences in FeNO level between urban and suburban? as this should show the overall differences.
- FIGURE 1: ---- it is not clear, and there is a missing text.
- Line 207-213: ---- This part seems related to "Discussion" not a result. The author should revise this paragraph.
- TABLE 3: ---- Not clear what is the role/justification of the "Reference" column? More clarification is needed here.
- TABLE 4: ---- The "OR" should be removed from the "table title".
Discussion:
- The 1st paragraph should summarize the overall study findings.
- The limitation section needs to consider other factors such as:
self-reporting asthma
No. of tested allergens.
Conclusions:
- The conclusion is not comprehensive, and I would suggest rewriting for more clarification and considering the overall recommendation and clinical implications.
Author Response

(The authors gave the same response as above.)

Round 2
Reviewer 1 Report
I appreciate for authors’ willingness to incorporate suggestions. I have a few following suggestions for further clarification.
Methods 2.3
I suggest to provide the hours of measurements for each pollutant. Authors added the mismatch of the sampling hours possibly inducing exposure measurement error as a limitation to the Discussion. However, it is worthwhile to provide in the Methods as this is the key characteristics of this study.
Related to sampling hours of exposure, it will be also helpful to clarify the hours of outcome measurements. Authors mentioned “the clinical assessment and indoor air monitoring were carried out at the same time from August until November 2018 and in early February 2019". Does this mean that FeNO measurements were obtained for the same hours as indoor air pollution? However, even sampling hours of air pollution varied, as authors mentioned in the Discussion. Different hours of outcome measurements such as morning and afternoon may affect findings. Please clarify the hours of exposure and outcome.
Discussion
In the first sentence, authors mentioned “this comparative cross-sectional study explored the association of FeNO levels with indoor PM2.5, PM10 and NO2 exposure”. However, authors actually investigated all criterial pollutants and found the associations for these three. Please clarify.
Author Response
We thank the reviewer for your valuable input and comments on the manuscript. We hope that our reply to each of your comments/suggestions finds your approval. We additionally uploaded the response letter as a file.

Reviewer 2 Report
The authors have addressed my comments very well as required, and now it is clear for me.
Thank you.
Author Response
Thank you for your constructive comments.
